# A new stem sarcopterygian illuminates patterns of character evolution in early bony fishes

Jing Lu[1], Sam Giles[2], Matt Friedman [2,3] & Min Zhu [1]

Discoveries of putative stem sarcopterygians from the late Silurian and Early Devonian of South China have increased our knowledge of the initial diversification of osteichthyans while also highlighting incongruities in character evolution in this major jawed vertebrate group. Character-rich endocrania are incompletely preserved for early bony fishes, limiting a detailed understanding of complex internal morphology and evolutionary changes in the cranium. Here we report a new sarcopterygian (*Ptyctolepis brachynotus* gen. et sp. nov.) from the Pragian (Early Devonian) of South China, which preserves a unique example of a completely ossified otoccipital division of the braincase in a stem lobe-finned fish. The hyomandibular facets are paired but lie dorsal to the jugular canal, representing a hitherto unobserved combination of derived and primitive character states. This new taxon prompts a reassessment of early osteichthyan interrelationships, including the phylogenetic placement of psarolepids, which might branch from the osteichthyan—rather than sarcopterygian—stem.

[1] Key Laboratory of Vertebrate Evolution and Human Origins of Chinese Academy of Sciences, Institute of Vertebrate Paleontology and Paleoanthropology, Chinese Academy of Sciences, Beijing, 100044, China. [2] Department of Earth Sciences, University of Oxford, South Parks Road, Oxford, OX1 3AN, UK. [3] Museum of Paleontology and Department of Earth and Environmental Sciences, University of Michigan, 1109 Geddes Ave, Ann Arbor, MI 48109, USA. Correspondence and requests for materials should be addressed to J.L. (email: lujing@ivpp.ac.cn) or to M.Z. (email: zhumin@ivpp.ac.cn)

Recent discoveries of osteichthyans (bony vertebrates comprising bony fishes and tetrapods) from the Ludfordian (Ludlow, late Silurian; ~425 million years ago, mya) to Lochkovian (Early Devonian; ~415 mya) have highlighted the significance of South China in understanding the diversification of lobe-finned fishes[1–5], as well as actinopterygians[6] and gnathostomes[7–9] more generally. Chief among these are the psarolepids (sensu ref. [10]), a clade of apparent stem sarcopterygians known from articulated (Guiyu[1] and Sparalepis[10]), dissociated postcranial and cranial (Psarolepis[5]) and cranial only (Achoania[4]) remains. Despite early phylogenetic ambiguities that included associations with porolepiforms[11] and the osteichthyan stem[5,12], placement of psarolepids as stem lobe-fins—and thus crown bony fishes—has become one of the dominant motifs of systematic analyses of early vertebrates[1–3,5,6,8,10,12–17]. This family of results posits a well-populated sarcopterygian stem peppered with incongruous character transformations. However, a steady stream of discoveries revealing unexpectedly plesiomorphic aspects of psarolepid anatomy, including the presence of 'placoderm'-like pelvic fin girdles and generalized dental histology[18,19], have amplified apparent complexities in early osteichthyan evolution. Association of these taxa with the sarcopterygian stem nevertheless persists.

While most recent work on the early diversification of bony fishes has centered on material from deposits flanking the Silurian–Devonian boundary, including those yielding

psarolepids, slightly younger strata have also provided important new fossils. Key among these is the Posongchong Formation of South China, which contains an abundance of osteichthyans including a diverse array of crown sarcopterygians, such as the earliest tetrapodomorphs[20], anatomically modern coelacanths[21], and early onychodonts[22,23]. Significantly, these fossils are all well preserved with character-rich endocrania.

Here we report a new sarcopterygian from the Posongchong Formation (~409 mya, Pragian, Early Devonian) of Yunnan, China. This taxon is represented by a single well preserved and completely ossified otoccipital division of the skull measuring 4.4 cm in width, 1.8 cm in length, and 2.7 cm in height, which suggests an individual considerably larger than other known members of the co-occurring sarcopterygian fauna (e.g. Tungsenia[20] and Euporosteus yunnanensis[21]). This region of the braincase is poorly known in sarcopterygians of Pragian age or older, with the only complete examples found in Styloichthys[3] and representatives of Dipnomorpha (Youngolepis[24] and Powichthys[25]), all of which are interpreted as crown sarcopterygians. The new taxon combines an anteroposteriorly short postparietal shield with vermiform ornament, while the braincase lacks vestibular fontanels and shows an unprecedented condition of the hyomandibular facet. The intact braincase permits reconstruction of a full endocast of the otic region, representing the second oldest complete osteichthyan example after Youngolepis[24]. Based on these and other observations, we include this new taxon in a

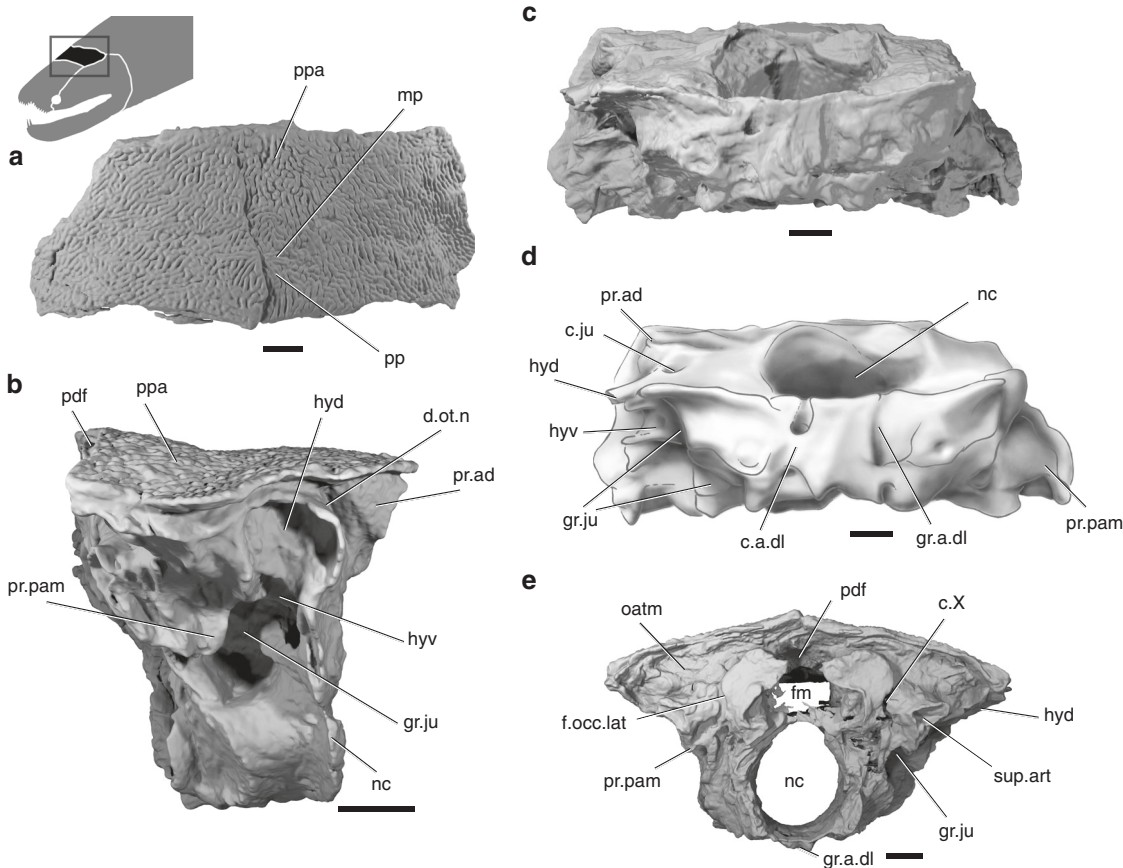

**Fig. 1** The skull of *Ptyctolepis brachynotus* gen. et sp. nov. (IVPP V23386). High-resolution CT rendering of specimen in dorsal (**a**), right lateral (**b**), ventral (**c**), and posterior (**e**) views and (**d**) interpretive drawing in ventral view. c.a.dl canal for lateral dorsal aorta, c.X canal for vagus nerve, d.ot.n dorsal branch of the otic lateral line nerve, fm foramen magnum, f.occ.lat lateral occipital fissure, gr.a.dl groove for lateral dorsal aorta, gr.ju groove for jugular canal, hyd dorsal hyoid articular area, hyv ventral hyoid articular area, mp middle pit-line, nc notochordal canal, oatm attachment area for trunk musculature, pdf posterior dorsal fontanel, pp posterior pit-line, ppa postparietal, pr.ad antero-dorsal process, pr.pam parampullary process, sup.art articular area for suprapharyngobranchial. Scale bar, 5 mm

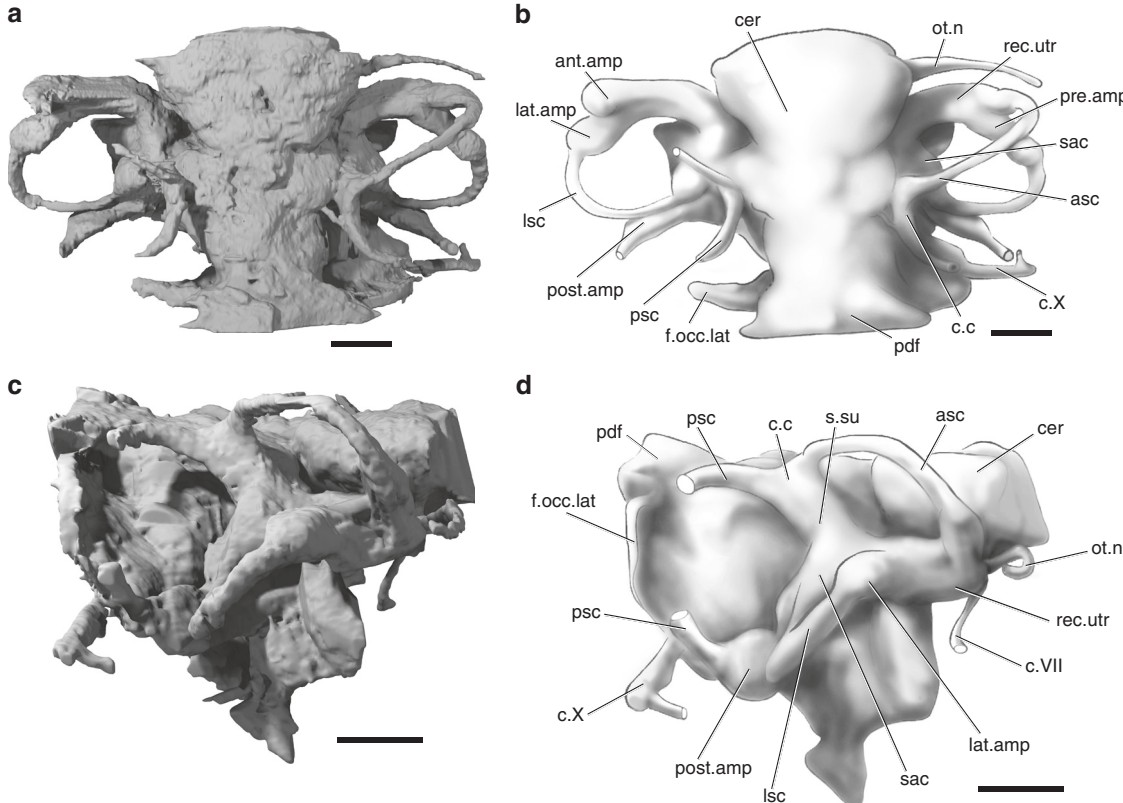

**Fig. 2** Digital neurocranial endocast of *Ptyctolepis brachynotus* gen. et sp. nov. **a** Rendering and **b** interpretive drawing in dorsal view; **c** Rendering and **d** interpretive drawing in right lateral view. ant.amp ampulla of anterior semicircular canal, asc anterior semicircular canal, cer cerebellum, c.c crus commune, c.VII facial nerve, c.X vagus nerve, f.occ.lat lateral occipital fissure, lat.amp ampulla of lateral semicircular canal, lsc lateral semicircular canal, ot. n otic lateral line nerve, pdf posterior dorsal fontanel, post.amp ampulla of posterior semicircular canal, pre.amp preampullary canal, psc posterior semicircular canal, rec.utr utricular recess, sac sacculus, s.su sinus superior. Scale bar, 5 mm

revised cladistic analysis to infer its phylogenetic position and evaluate its impact on the pattern of branching and character evolution deep within osteichthyan phylogeny.

## Results

### Systematic paleontology.

<div align="center">

Osteichthyes Huxley, 1880[26]

Sarcopterygii Romer, 1955[27]

*Ptyctolepis brachynotus* gen. et sp. nov.

</div>

**Etymology**. Generic name referring to vermiculate-ridge ornamentation on the skull roof, from Greek *ptyktos* (fold) and *lepidos* (scale). Specific name from Greek *brachyno* (shorten) and *otos* (ear, otic region), meaning the short otic region.

**Holotype**. IVPP V23386, a complete posterior cranial portion of the skull, Institute of Vertebrate Paleontology and Paleoanthropology (IVPP), Chinese Academy of Sciences (CAS), Beijing, China.

**Locality and horizon**. Outcrop near the Qingmen Reservoir in the suburb of Zhaotong, northeastern Yunnan. The fossil horizon belongs to the Posongchong Formation, which mainly comprises yellowish sandstones. In addition to lingulid brachiopods and plant remains, the associated biota includes an abundance of galeaspid agnathans[28–31], 'placoderms'[32], and osteichthyans[20–23,31,33,34]. Specimens from this horizon are three-dimensionally preserved with little distortion. The age of the Posongchong Formation is considered to be late Pragian, mainly based on the correlation of marine invertebrates and conodonts from the overlying Pojiao Formation[35,36].

**Diagnosis**. A sarcopterygian characterized by the unique combination of: laterally broad and anteroposteriorly short postparietal shield, very large notochordal canal, paired hyomandibular facets dorsal to jugular canal, basicranial fenestra and vestibular fontanel both absent, middle and posterior pit lines lying close to midline of skull.

**Description**. The skull roof is represented by the posterior cranial portion (otoccipital shield/postparietal shield; Fig. 1 and Supplementary Figs. 1 and 2). The shield is much wider than long, with a width:length ratio (~2.5) much higher than that of *Psarolepis* (~1.8) or *Guiyu* (~1.3). The anterior margin of the postparietals (see ppa in Fig. 1a) is straight, suggesting that a dermal cranial joint was well developed. The ornament comprises short, vermiform ridges (Fig. 1 and Supplementary Figs. 1 and 3) most similar to those of *Guiyu*[1] or '*Ligulalepis*'[13], rather than the smooth, porous ornament seen in *Psarolepis*[37] and *Meemannia*[6]. However, CT data show small pores open between the ridges (Supplementary Fig. 3). The midline suture between the postparietals is clear, with the left bone shifted slightly over the right (Fig. 1a and Supplementary Fig. 1). The middle and posterior pit lines are situated posteriorly and meet at the midline (see mp and pp in Fig. 1a), as in *Janusiscus*[16], *Guiyu*, *Dialipina*[38], and '*Ligulalepis*,' but unlike the more widely spaced pit lines of *Meemannia* and *Psarolepis*. There is no indication of the anterior pit-line on the postparietals, which therefore must have been borne on the parietals. The main lateral line canal extends anteroposteriorly

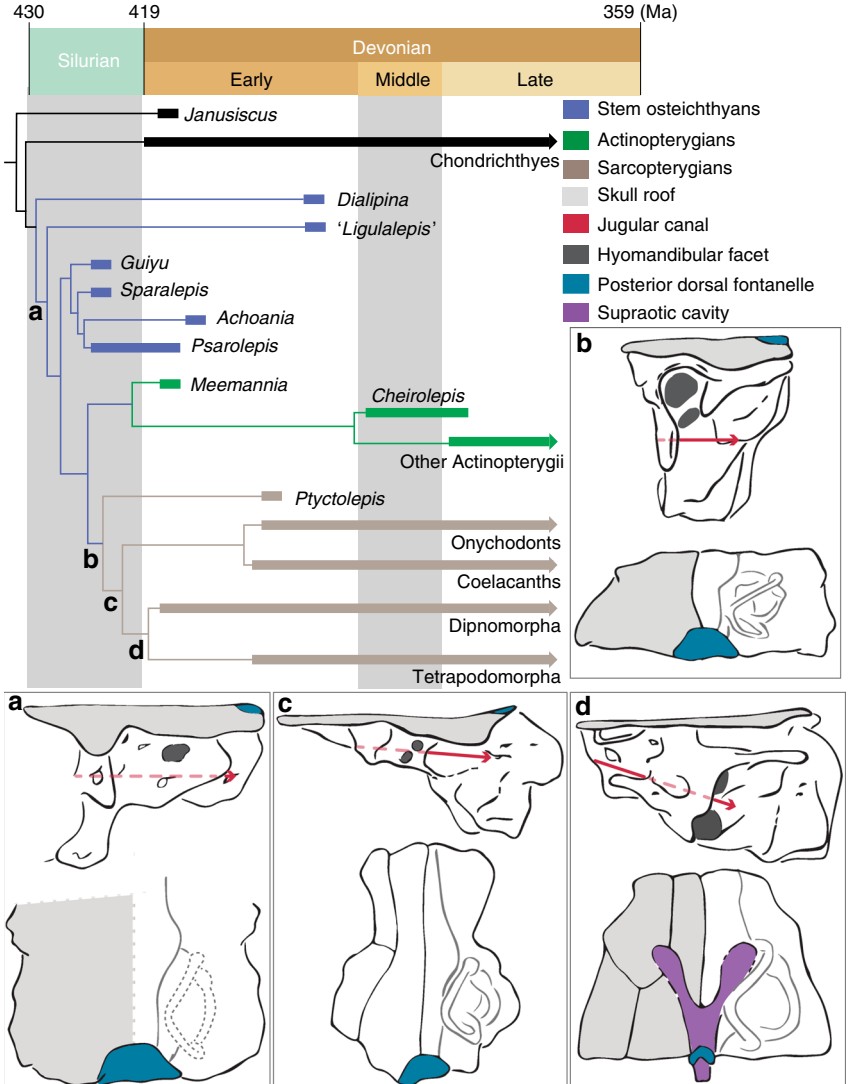

**Fig. 3** Simplified phylogeny showing braincase evolution in osteichthyans. **a** Single hyomandibular facet dorsal to the jugular vein, as shown in 'Ligulalepis'[13]. **b** Double hyomandibular facet dorsal to the jugular vein, as shown in Ptyctolepis. **c** Double hyomandibular facet straddling the jugular vein, as shown in Qingmenodus[22]. **d** Double hyomandibular facet dorsal to the jugular vein; supraotic cavity present, as shown in Eusthenopteron[40]. Not drawn to scale

near the lateral margin of the postparietal (see soc in Supplementary Fig. 2a). Sutures between the postparietal and more lateral bones (i.e. tabular and supratemporal) cannot not be traced.

The otoccipital division of the braincase is well ossified (Fig. 1 and Supplementary Figs. 1 and 2). The most conspicuous feature of the neurocranium is the large notochordal opening, which is fully two-thirds of the height of the occipital region (see nc in Fig. 1e and Supplementary Fig. 2b).

There is no well-developed otic shelf of the kind seen in coelacanths, onychodonts, and tetrapodomorph fishes[23,39,40], but there is a modest ridge that extends along the ventral margin of the jugular groove in a corresponding position (Fig. 1c–d). The dorsal margin of the jugular groove is marked by a suprajugular ridge, similar to that in Youngolepis[24]. As in coelacanths, onychodonts, and Psarolepis[11,39,41], the trigeminal nerve does not pass through the otoccipital. A small antero-dorsal process, similar to that of Psarolepis[11] is present on the right side (see pr. ad in Fig. 1 and Supplementary Figs. 1 and 2), but missing from the damaged left side. Posteriorly, the transverse otic process (cf. Cheirolepis[42], sometimes referred to as the lateral commissure; for a discussion of terminology of lateral processes of the braincase in

early gnathostomes, see ref. [16]) is pierced by the jugular canal and bears two facets for the hyomandibula (see hyd and hyv in Fig. 1 and Supplementary Figs. 1 and 2), as in other sarcopterygians (e.g. Qingmenodus[22], Youngolepis[24], and Styloichthys[3]). The laterally facing dorsal facet is larger, and is separated from the ventrally directed lower facet by a thin ridge of bone. This contrasts with the single hyomandibular facet present in Psarolepis[11], 'Ligulalepis'[13], and actinopterygians[43]. Unlike other taxa with paired facets, both articular areas lie dorsal to the jugular vein rather than straddling it. The otoccipital fissure, through which the vagus nerve exits, is well developed (see c.X and f.occ.lat in Fig. 1e), and a basicranial fenestra is absent, as in Styloichthys[3] and Youngolepis[24]. Ptyctolepis lacks a vestibular fontanel, like Styloichthys, Qingmenodus, and coelacanths[14].

An opening on the posterodorsal surface of the occiput (see pdf in Fig. 1e), continuous with the foramen magnum but set off from it by a constriction, corresponds to the posterior dorsal fontanel. The articular area for the first suprapharyngobranchial is borne on a stout post-otic process at the level of the foramen magnum (see sup.art in Fig. 1e). A depression dorsal to the post-otic process likely served as an attachment area for trunk musculature (see otam in Fig. 1), but is not divided into distinct regions (cf.

*Youngolepis*, and, to a much greater degree, *Eusthenopteron*). The slot-shaped foramen magnum (see fm in Fig. 1e) resembles that of *Diplocercides* (incorrectly labeled as an endolymphatic opening previously)[40], and is separated from the large notochordal canal by a partially mineralized shelf. In ventral view, the parachordal region bears two parallel grooves for the lateral dorsal aortae (see c.a.dl and gr.a.dl in Fig. 1 and Supplementary Figs. 1 and 2), which would have joined posterior to the occiput. The left lateral aorta was entirely extramural, but the right groove is partially enclosed by a thin sheet of bone.

The internal surface of the otoccipital of *Ptyctolepis* is well mineralized, allowing a cranial endocast to be produced (Fig. 2 and Supplementary Fig. 4). The endocast corresponds to the hindbrain (rhombencephalon), inner ear, and associated structures. The cerebellum (see cer in Fig. 2 and Supplementary Fig. 4) is well developed, although it appears to lack cerebellar auricles. Swellings situated posterior to the widest part of the hindbrain are associated with the sinus superior rather than the cerebellum, and may have partially accommodated the endolymphatic ducts within the cranial cavity (Fig. 2a). Two canals extend from the lateral face of the cerebellum, near its ventral margin (see ot.n and c.VII in Fig. 2 and Supplementary Fig. 4). One curves laterally and slightly posteriorly and corresponds to the otic lateral line nerve. The other runs posteroventrally and probably transmitted the facial nerve. The labyrinth cavity is well preserved, with three pairs of semicircular canals (anterior, posterior, and lateral), spaces for ampullae, the crus commune, sinus superior, and part of the sacculus clearly visible (see asc, psc, lsc, ant.amp, pre.amp, lat.amp, post.amp, c.c, s.su, and sac in Fig. 2 and Supplementary Fig. 4). The utricular recess is laterally elongate (see rec.utr in Fig. 2), resulting in a large separation between the ampulla of the anterior semicircular canal and the brain cavity. Thus, the anterior semicircular canal is oriented at ~45° to the brain cavity (see asc in Fig. 2 and Supplementary Fig. 4). This contrasts with a more typical angle of no more than 30° seen in almost all other early crown gnathostomes (e.g. chondrichthyans[44,45], actinopterygians[46], and sarcopterygians[24,40]). The semicircular canals are very narrow in section (Supplementary Fig. 5), about half the width of the ampullae. The anterior and posterior semicircular canals join in a common crus that extends some way above the dorsal roof of the cranial cavity (see psc and cc in Fig. 2 and Supplementary Fig. 4), as in other sarcopterygians[22], chondrichthyans[45], and primitive actinopterygians[46]. A preampullary canal (see pre.amp in Fig. 2a) separates the ampulla of the lateral canal from the utriculus. In addition, a swelling of the lateral canal as it rejoins the cranial cavity posteriorly gives the impression of a second ampulla (Fig. 2), as in *Youngolepis*[24]. The sacculus is only partially preserved (see sac in Fig. 2 and Supplementary Fig. 4), but appears to have limited lateral extent as in actinopterygians, *Youngolepis*, and *Eusthenopteron*[24,40,46], rather than being laterally bulbous as in *Qingmenodus* and coelacanths[22,40]. The path of the posterior semicircular canal is incompletely preserved, but it rejoins the cranial cavity via its ampulla ventral to the lateral canal.

**Phylogenetic results**. Our parsimony analysis recovers 861,680 trees with a length of 802 steps (character optimization for a single MPT given in Supplementary Data 1). *Ptyctolepis* is consistently resolved as a stem sarcopterygian (Bremer decay index = 3; Fig. 3 and Supplementary Fig. 6), and shares with the crown group the following three characters: spiracular groove absent from transverse otic process; double hyoid arch articulation on braincase; parachordals mediolaterally constricted relative to the otic capsules. The crown itself is supported by two synapomorphies, the latter of which has a CI of 1: basicranial fenestra;

hyomandibular articulation straddles jugular canal. In marked contrast to almost all previous phylogenetic analyses[1–6,8,10,13–17,47] (but see refs. [5,12]), psarolepids (i.e. *Guiyu*, *Sparalepis*, *Achoania*, and *Psarolepis*)[10] form a clade on the osteichthyan, rather than sarcopterygian, stem, albeit with weak nodal support. This arrangement has been suggested through verbal argumentation, but not formal analysis, in some recent studies[10,18,19]. This clade (Bremer decay index = 3) is supported by four homoplastic characters: posterior flexion of dentary symphysis; internasal vacuities; extended prehypophysial portion of sphenoid absent; and pelvic fin spines. The characters supporting the osteichthyan crown node, and therefore excluding psarolepids, are: enamel(oid) on teeth (cf. ref. [19]); splint-shaped parasphenoid; elongate and tubular olfactory tracts; eyestalk attachment area absent; median dorsal plate absent (the latter with a CI of 1). The osteichthyan total group (Bremer decay index = 3; bootstrap support = 70%) shares the following synapomorphies: enamel(oid) present on dermal bones and scales; body scales with peg-and-socket articulation; body scales lacking a bulging base; teeth not ankylosed to dermal bones; and maxilla and dentary present. Other areas of our tree are broadly in agreement with past results[1,10,16,47], although most 'acanthodians' have collapsed into a series of polytomies. The high number of MPTs is largely the result of uncertainty in the branching position of 'acanthodians' (although all are resolved on the chondrichthyan stem in all MPTs). Tree support values are as follows: CI = 0.372; RI = 0.793; RC = 0.295.

The topology arising from Bayesian inference differs in that psarolepids are retained on the sarcopterygian stem (BPP = 0.95; Supplementary Fig. 7), in a 'conventional' position. In other respects, the two analyses recover broadly similar results, with the exception of minor discrepancies in relationships among 'placoderms' and 'acanthodians,' and the placement of *Ramirosuarezia* (with fairly low support; BPP = 0.65) as a stem chondrichthyan rather than stem gnathostome.

## Discussion

Osteichthyans are well represented in the Early Devonian, but intact braincases for members of the group are rarely preserved intact. This is particularly apparent for the otoccipital region, where continuous otoccipital and ventral otic fissures, often in conjunction with a vestibular fontanel, seems to result in frequent dissociation of the posterior and ventral portions of the braincase (e.g. 'Ligulalepis'[13], *Meemannia*[6], *Psarolepis*[11], and *Achoania*[4]). Furthermore, in taxa with a jointed cranium, the posterior half (postparietals plus otoccipital regions) is poorly represented in dissociated material, as preservation appears biased toward the ethmoid shield and associated endoskeleton (e.g. *Powichthys*[48], *Diabolepis*[49], and *Tungsenia*[20]). By providing the first well-preserved, complete otoccipital division of a stem sarcopterygian, *Ptyctolepis* allows us to revisit endocranial character evolution in early sarcopterygians, particularly with respect to the supraotic cavity and the relative positions of the hyomandibular articulation and jugular canal. When taken in conjunction with recent reports of articulated material[1,10], these findings present an opportunity to reconsider early osteichthyan anatomy as a whole.

The posterior dorsal fontanel is a median opening on the dorsal surface of the endocranium, and in taxa with a macromeric dermal skeleton it lies close to the posterior margin of the skull roof (Fig. 3). It is absent in stem gnathostomes (e.g. *Dicksonosteus*[50] and *Janusiscus*[16]), but present in chondrichthyans inclusive of acanthodians (*Acanthodes*[15]; *Cladodoides*[45], and *Pucapampella*[47]) and likely stem osteichthyans ('Ligulalepis'), and is a probable synapomorphy of crown gnathostomes (Supplementary Data 1). Faint grooves on the visceral surface of the endocranial

roof in 'Ligulalepis' lead from the sinus superior to the posterior dorsal fontanel[51], suggesting that the endolympatic ducts were transmitted within the dorsal part of the cranial cavity (Fig. 3a). Unlike in chondrichthyans[45], the posterior dorsal fontanel is confluent with the otoccipital fissure. The primitive osteichthyan condition, with a semicircular posterior dorsal fontanel that is continuous with the fissure, and with endolymphatic dusts transmitted through the cranial cavity, is also characteristic of early sarcopterygians based on our new evidence from Ptyctolepis (Fig. 3b). The fontanel becomes progressively larger in actinopterygians, eventually extending from the otoccipital fissure to the level of the sinus superior (e.g. Mimipiscis[46]), but the condition in crown sarcopterygians is somewhat different. In Youngolepis, an opening corresponding to the posterior dorsal fontanel (although its homology has previously been left equivocal[24,40]) leads into the cranial cavity and is confluent with the fissure, as in other early osteichthyans. In addition, this is joined by a deep ridge on the roof of the cranial cavity that continues anteriorly and terminates as a blind-ending canal anterior to the sinus superior. Posteriorly, it extends toward the rear of the braincase, meeting its fellow at the midline just anterior to the posterior dorsal fontanel, with which it is continuous. This structure is referred to as the supraotic cavity, and assumed to house the endolymphatic sac. This is elaborated further in taxa such as Eusthenopteron (Fig. 3d), Gogonasus, and lungfishes, where the supraotic cavity is largely or entirely separate from the cranial cavity after leaving the sinus superior[40,52], and is not continuous with the otoccipital fissure. A structure apparently corresponding to the posterior dorsal fontanel is also present, but does not appear to connect to the otoccipital fissure. No such division between the supraotic and cranial cavities is present in Psarolepis, onychodonts, coelacanths, or Ptyctolepis. Comparison with outgroups such as actinopterygians (e.g. Mimipiscis[46]) and chondrichthyans (e.g. Cobelodus[45]) suggests that this is the primitive condition.

The structure of the hyomandibular facet in osteichthyans falls broadly into two categories: examples where a single facet is located dorsal to the course of the jugular vein, and examples where paired facets straddle the level of the jugular canal (Fig. 3). The first of these conditions appears primitive, and is found in the stem gnathostome Janusiscus, the probable stem osteichthyan 'Ligulalepis' (Fig. 3a), actinopterygians, and Psarolepis. The presence of paired facets bridging the jugular vein is classically considered a character of crown sarcopterygians (Fig. 3c–d), although in some members of this group there is a single facet thought to represent the coalescence of primitively paired articular areas (e.g. lungfishes). Ptyctolepis (Fig. 3b) presents a combination of these two contrasting arrangements. As in primitive osteichthyans, the hyomandibula articulates with the otic capsule dorsal to the jugular vein. However, it shows division of the facet into separate dorsal and ventral components, as in crown sarcopterygians. The placement of Ptyctolepis in our phylogenetic analysis as a member of the sarcopterygian stem lineage suggests that division of the hyomandibular facet preceded the ventral extension of the hyoid articulation across the jugular vein canal.

The unusual character combinations of psarolepids have been clear since their initial discovery and description[5,11], with more recent discoveries of articulated material - only serving to magnify their mosaic bodyplans. Central among the issues raised by psarolepids are features present in the group and osteichthyan outgroups, but absent in other definitive crown bony fishes: an eyestalk attachment area (the putative eyestalk in Styloichthys is resolved as non-homologous in our analysis), a median dorsal plate, dorsal-fin spines, pectoral-fin spines, pelvic fin spines, and dermal pelvic girdles. Previous phylogenetic consensus—

anchored by the presence of classic sarcopterygian features like an intracranial joint and cosmine[53]—regarded psarolepids as stem lobe-finned fishes, demanding one of two possible evolutionary histories for each of these characters: retention of the primitive condition in psarolepids with parallel loss in actinopterygians and other sarcopterygians, or absence in the common ancestor of sarcopterygians and actinopterygians with a reversal in psarolepids. While the integrity of the intracranial joint as a sarcopterygian feature remained intact, other evidence for a lobe-finned—or even crown osteichthyan—affinity of psarolepids has eroded with further scrutiny. First is the report that unlike actinopterygians and sarcopterygians, psarolepid teeth lack enamel[54]. Second is the discovery that many of the individual traits that characterize the complex tissue cosmine are present in probable (Meemannia) and definitive (Cheirolepis) actinopterygians[6], meaning that dermal bone histology of psarolepids is not compelling evidence of sarcopterygian affinity. These and other observations of primitive aspects of psarolepid anatomy have triggered a steady stream of discussion that these taxa might be stem osteichthyans rather than early sarcopterygians[10,18,19], contrary to the apparent consensus arising from formal analyses.

Placement of psarolepids as stem osteichthyans provides a potential solution to the unusually high number of generalized features present in this group. Median dorsal plates, for example would now represent a symplesiomorphy inherited from stem gnathostomes and lost at the osteichthyan crown node, while tooth enamel is a crown osteichthyan synapomorphy. Some issues, however, remain. Tooth whorls are present in chondrichthyans (inclusive of 'acanthodians'), psarolepids, onychodonts, Gavinia, Styloichthys, and porolepiforms, but due to absences at nodes preceding or subtending the gnathostome (i.e. Entelognathus), osteichthyan (actinopterygians), and sarcopterygian (some coelacanths and dipnomorphs) total groups, states for this character cannot be optimized deep in osteichthyan phylogeny. Multiple independent appearances are considered as parsimonious as a single appearance below the gnathostome crown node and several losses. Similarly, many characters pertaining to fin spines cannot be optimized as the condition is unknown in proximate stem gnathostome and stem osteichthyan taxa; fin spines are known to be absent only in the anatomically peculiar and poorly understood Dialipina[38].

In addition to these ambiguities resulting from missing data, a stem osteichthyan placement of psarolepids implies a more complicated evolutionary history for the intracranial joint than previously suspected. Rather than a synapomorphy of sarcopterygians, lost independently in tetrapods and lungfishes[53], it represents either convergence between psarolepids and sarcopterygians, or a character of psarolepids and crown osteichthyans, subsequently lost in actinopterygians and multiple sarcopterygian lineages. However, these conclusions remain tentative at best, as our Bayesian trees resolve psarolepids in the conventional stem sarcopterygian position with high nodal support; BPP = 0.95). Indeed, although the psarolepids are placed as stem osteichthyans in all most parsimonious trees, there is no significant difference in terms of tree length between this solution and one that places these taxa in a more conventional position on the sarcopterygian stem (Templeton test; $p = 0.8474$). We anticipate that further discoveries of early osteichthyan material from the Silurian of China, and additional study of existing fossils, will help to clarify the evolutionary histories of these characters, as well as providing critical tests of competing hypotheses for the placement of psarolepids. More definitive resolution of this problem will have consequences not only for our understanding of character evolution among early osteichthyans, but also the timescale of vertebrate diversification, as the psarolepid Guiyu has become a

key fossil marker in molecular clock studies[55]. In the light of present ambiguities, we regard the placement of these taxa as uncertain, but limited to either the osteichthyan or sarcopterygian stem.

## Methods

**High-resolution computed tomography.** The holotype of *Ptyctolepis brachynotus* gen. et sp. nov. (IVPP V23386) was scanned at the Institute of Vertebrate Paleontology and Paleoanthropology (IVPP), Chinese Academy of Sciences (CAS), Beijing, China, using 225 kV microCT (developed by the Institute of High Energy Physics, CAS). The specimen was scanned with a beam energy of 130 kV and a flux of 100 mA at a detector resolution of 27.4 μm per pixel, using a 1440° rotation with a step size of 0.25° and an unfiltered aluminum reflection target. A total of 1440 transmission images were reconstructed in a 2048 × 2048 matrix of 1536 slices. Scan data were analyzed using Mimics v.18.01 (http://biomedical.materialise.com/mimics; Materialize) and imaged in Blender (blender.org).

**Phylogenetic data set assembly and analyses.** Our data set is modified from ref. [6] (which is largely based on ref. [16]). It has been expanded by the addition of four taxa (*Ptyctolepis*, *Achoania*, *Qingmenodus*, and *Sparalepis*) and nine characters (both novel and taken from the literature[10,56], giving a total of 278 characters and 94 taxa (see Supplementary Table 1 and Supplementary Note 1. Data matrix given in Supplementary Data 2). Codings for some taxa have been updated in light of recent publications (e.g. *Doliodus*[57]; *Romundina*[58]; and *Psarolepis*[19]) and to correct previous miscodes. We have also reformulated character 186, relating to the presence of median dorsals, to reflect that dermal plates are coded as inapplicable (rather than absent) in 'acanthodians' and chondrichthyans. An equally weighted parsimony analysis (with 500 random addition sequences, five trees held at each step, maxtrees set to automatically increase, nchuck = 10,000, chuckscore = 1) was performed in PAUP* 4.0a150[59], with the outgroup constrained as [Galeaspida [Osteostraci[ingroup]]]. Bootstrap values were calculated in PAUP using 500 replicates of a heuristic search, with five trees held at each step, rearrlimit = 50,000,000, limitperrep = yes, nchuck = 10,000, chuckscore = 1. Bremer decay values were also calculated in PAUP. Bayesian analysis was run under the Mkv model in MrBayes v.3.2.6[60], until the standard deviation of split frequencies reached less than 0.01, indicating convergence had been reached. The first half of each run was discarded as burnin.

**Nomenclatural acts.** This published work and the nomenclatural acts it contains have been registered in ZooBank, the proposed online registration system for the International Code of Zoological Nomenclature (ICZN). The ZooBank LSIDs (Life Science Identifiers) can be resolved and the associated information viewed through any standard web browser by appending the LSID to the prefix "http://zoobank.org/." The LSIDs for this publication are: urn:lsid:zoobank.org:pub:C1D7AD35-DEC1-4F6D-B0C8-D94DEB89F710 (article); urn:lsid:zoobank.org:act:5CBD2D52-0E50-441E-8BD4-74D9204E3570 (genus); and urn:lsid:zoobank.org:pub:C1D7AD35-DEC1-4F6D-B0C8-D94DEB89F710 (species).

**Data availability.** The CT data that support the findings of this study, as well as 3D surface files of described material, are available in figshare[61] with the identifier https://doi.org/10.6084/m9.figshare.5458165. All other data files are included in the Supplementary Information.

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

## Acknowledgments

We thank Y.-M. Hou for assistance with CT scanning and C.-H. Xiong for specimen preparation. This work was supported by the National Natural Science Foundation of China (41472016, 41530102), Key Research Program of Frontier Sciences of CAS (QYZDJ-SSW-DQC002), and Strategic Priority Research Program of CAS (XDPB05) to J.L. and M.Z., a Junior Research Fellowship from Christ Church, Oxford, and a L'Oréal-UNESCO For Women in Science Fellowship, both to S.G., and the Philip Leverhulme Prize and John Fell Fund, both to M.F.

## Author contributions

The project was conceived by J.L. and M.Z. All authors performed the research. M.Z. and J.L. carried out field work. S.G., J.L., and M.F. generated the CT renderings. Figures were produced by J.L. and S.G. S.G., M.F., and J.L. conducted the phylogenetic analyses. All authors participated in the interpretation of the specimen data and writing the manuscript.

## Additional information

**Competing interests:** The authors declare no competing financial interests.

