## [Peer Review File · Nature Communications]

Reviewers' comments:

Reviewer #1 (Remarks to the Author):

This is a really interesting paper describing a unique and important contribution to the study of the origin of the bony vertebrate clade and of its early members. Quite remarkable material is coming out of China, and this paper describes the latest. As with the earlier discoveries, this one is showing the picture to be yet more complex, with individual taxa possessing different mixes of primitive and derived characters. The analysis here interprets the new specimen as a stem sarcopterygian, but the specimen also shows some similarities to actinopterygians, as you might expect from such a basal taxon. The analysis also suggests that *Psarolepis* is more likely a basal osteichthyan than a basal sarcopterygian, which seems to make sense of some of its conflicting characters.

I note that there does not seem to be any evidence of an occipital arch in this braincase specimen, except for a possible basioccipital. This means that the lateral otic fissure completely separates the otoccipital from the occipital arch. Where would the vagus nerve exit?

I have very little to comment on except for the fact that the lateral semicircular canal is labelled 'external' in the figures and 'horizontal' in the text. Lateral is the more usual term in current auditory literature. Also in the figures '*Ligulalepis*' is in quotation marks, but it is not so in the text. What is the reason for this, and should it not be made consistent?

Reviewer #2 (Remarks to the Author):

This is a well researched and well written manuscript. Whilst it could be considered just another description and analysis of a partial fossil, the results from South China in recent decades have proved to be the most significant vertebrate fossil discoveries for the early part of our evolutionary history for perhaps the last century. For some reason that place was an 'evolutionary cradle' for all gnathostomes (jawed vertebrates), with numerous taxa either older than, or more primitive than (or both) comparable fossils from anywhere else in the world.

The new taxon *Ptyctolepis* is yet another example, and like any new fossil, its significance depends on a competent analysis of its morphology in the context of current competing hypotheses. This has been very adequately done in this paper in my opinion, with clear description of new features and character combinations, followed by a full consideration of the implications for interpretation of morphology, and phylogenetic placement, for other key taxa from China, many of which have been described in *Nature*.

Thus there is no problem to accept the major claim of the paper, that it 'illuminates patterns of character evolution in early bony fishes'. It will certainly be of interest to, and influence thinking within, the field of early vertebrate studies.

This paper is also likely to have wider influence, because it concerns the character combinations that define not only various extinct groups, but the two most diverse groups

of living vertebrates, the Actinopterygii, dominating the modern aquatic environment, and the ancestral lobe-fins [sarcopterygians], and all their tetrapod descendants in the modern terrestrial environment.

Generally this MS is very well written and clearly expressed. The figures are excellent and necessary, and the MS cannot be shortened without loss of important information. Nevertheless, I suggest the authors consider various points where I have small queries, or can suggest clarification of the text. These are marked on the attached pdf, and listed below. So I would propose perhaps that these minor revisions be done, subject to editorial advice. However none is important enough to detract from the high quality of this submission, which in my assessment would be acceptable for publication in its current form.

Detailed queries/comments [see annotated pdf]:

lines 20-22. perhaps one or other 'character-rich' could be replaced by other words [sounds a bit repetitive]. I suggest some alternative words in line 22.

line 23. Other groups [e.g. psarolepids] have been regarded as stem sarcopterygians, even if analysis of this new taxon suggests otherwise, so better not to claim it is 'the first sarcopterygian' in the Abstract.

lines 61-62. Would be helpful to give more details of size estimates – just a few words in parentheses would do – e.g. other sarcopts about ?? cm total length compared with Ptyctolepis [total length estimated at about ?? cm].

lines 103-105. I get the impression from the figures that the holotype specimen may be very slightly distorted (i.e. not completely symmetrical, ignoring missing bits). If so this could slightly affect measurements and proportions - if sedimentary strata have been folded, this can stretch or compress fossils in different axes. Distortion would be evident in all the other fossils from the same locality, but I am not familiar with these. If this is relevant, perhaps a brief comment could be added under 'Locality and Horizon'.

lines 109-110. There are always openings in hard tissue between ornamental ridges or tubercles, so not sure if the CT data show evidence that these are 'pores' comparable to a pore-canal system.

lines 159-161. Perhaps two sentences here.

lines 197-201. The expression seems a bit cumbersome here. The phrase 'although ... in some recent studies' could be a separate sentence at the end, and the term 'verbal argumentation' would be better replaced with a few words perhaps expanding what the arguments actually are.

line 231. Any idea what may have caused that bias? i.e. preservation bias, collecting bias?

lines 271-272. I think this is better expressed as an assumption based on outgroup comparisons.

lines 312-314. I think more clearly expressed if the phrase 'contrary to ... analyses' is moved to the end; and authors to check where the three cited references are best placed.

Other

Minor typos/corrections/suggested improvements are also marked in lines 66, 132, 148, 211, 212, 242, 254, 324, 328, 331, 339, 341, 342, 348.

Reviewer #3 (Remarks to the Author):

The authors describe a partial braincase of a new Early Devonian osteichthyan. Although partial, the material is well preserved and allows recovery of characters related to the oto-occipital skeleton as well as an endocast of the hind brain and ears. The authors specifically identify several features that have bearing on the phylogenetic placement of this new taxon and implications for basal osteichthyan evolution more generally. One key interpretation is that the new taxon is a stem sarcopterygian based on three characters shared with crown sarcopterygians: 1) spiracular groove absent from lateral commissure; 2) double hyoid arch articulation on braincase; and 3) parachordals mediolaterally constricted relative to otic capsules. The most important conclusions of the manuscript relate to the new tree presented in Figure 3, which differs from previous interpretations in formally placing forms such as †Guiyu and †Psarolepis as stem osteichthyans, †Meemania as a stem actinopterygian, and the new taxon as a stem sarcopterygian. If this interpretation holds up, it will substantially contribute to our interpretation of basal osteichthyan evolution.

My comments on the manuscript are minor and editorial, but I would like to suggest two style points that would improve general readability of this (and other) papers in paleontology published in Nature Communications. First, I am a strong proponent of the dagger (†) symbol because it instantly clarifies to readers that the authors are referring to an extinct taxon, about which we will always know less than an extant taxon. It also helps clarify which named higher groups are extant, and this is particularly important in any tree that combines extinct and extant taxa as in Figure 3. Second, in figure callouts that refer to specific anatomical structures, it is a great help to readers if the abbreviation to the feature can be included as part of the callout. In reviewing this manuscript, I found several instances where an anatomical feature mentioned in a callout was difficult to find on the figure itself, and note some instances of this in my detailed comments below. Now I realize that my quibbles about these two style points may trace to specific style preferences of Nature Communications; but if so, then I would like to see a change in the journal's style standards to accommodate these preferences at an author's discretion.

Line 20 – “this dominant vertebrate group” antecedent to “this” could be unclear; also, what exactly do you mean by “dominant?” Presumably speciose – but this is so self evident that I am not sure this phrase helps your story.

Line 20 and Line 21 – Use of the phrase “character-rich endocrania” twice in two lines reads like a hard sell. Once would be enough.

Line 46 – Perhaps replace “primitive” with “plesiomorphic”

Line 59 – Replace “This taxon is only represented” with “This taxon is represented”

Line 66 – Replace “sarcopterygian” with “sarcopterygians”

Line 67 – Replace “while the braincase shows an absence of vestibular fontanelles and an unprecedented condition of the hyomandibular facet” with “while the braincase lacks

vestibular fontanelles and shows an unprecedented condition of the hyomandibular facet.

Line 84 – Replace “a complete posterior cranial portion” with “a complete posterior cranial portion of the skull”

Lines 102-103 – Replace “The skull roof is represented only by the posterior cranial portion” with “The skull roof is represented by the posterior cranial portion of the skull”

Line 120 – Is the callout to Figure 2 correct here? Does not seem relevant to the rest of the paragraph. Perhaps this should be Figure 1?

Lines 139-140 – This is an example where an indication to a specific label in the figure would help: “The otoccipital fissure, through which the vagus nerve exits, is well developed (Fig. 1e).” Where exactly is this in the figure?

Lines 145-146 – Again, I could not easily see that “articular area for the first supratharyngobranchial is marked by a stout post-otic process at the level of the foramen magnum (Fig. 1c-e).” A callout to a specific abbreviation would have helped.

Line 172 – Is the callout to Figure 1 correct? Perhaps Figure 2 shows this better?

Line 224 – Authors state that “intact braincases for members of the group are rarely completely preserved.” The sentence seems to imply that the new specimen has an intact braincase, but this specimen is not an intact braincase either, just the oto-occipital region.

Response to Reviewers' comments:

General:

We thank the three reviewers for their helpful comments that have allowed us to improve our contribution. They are enthusiastic both about the nature of anatomical interpretations, and the clarity with which they are figured in the main text and elsewhere. The suggestions are largely editorial, and we have made the changes to the manuscript requested.

Point-to-point response to the reviewers:

Reviewer #1 (Remarks to the Author):

This is a really interesting paper describing a unique and important contribution to the study of the origin of the bony vertebrate clade and of its early members. Quite remarkable material is coming out of China, and this paper describes the latest. As with the earlier discoveries, this one is showing the picture to be yet more complex, with individual taxa possessing different mixes of primitive and derived characters. The analysis here interprets the new specimen as a stem sarcopterygian, but the specimen also shows some similarities to actinopterygians, as you might expect from such a basal taxon. The analysis also suggests that Psarolepis is more likely a basal osteichthyan than a basal sarcopterygian, which seems to make sense of some of its conflicting characters.

I note that there does not seem to be any evidence of an occipital arch in this braincase specimen, except for a possible basioccipital. This means that the lateral otic fissure completely separates the otoccipital from the occipital arch. Where would the vagus nerve exit?

[Response]: The vagus nerve exits from the lateral occipital fissure (f.occ.lat in Figure 1). We added the label for the vagus nerve exit in Figure 1, and the corresponding text in the caption.

I have very little to comment on except for the fact that the lateral semicircular canal is labelled 'external' in the figures and 'horizontal' in the text. Lateral is the more usual term in current auditory literature.

[Response]: We modified all uses of 'external' and 'horizontal' semicircular canal to 'lateral semicircular canal' and the relevant abbreviations in both figure captions and figures.

Also in the figures 'Ligulalepis' is in quotation marks, but it is not so in the text. What is the reason for this, and should it not be made consistent?

[Response]: We refer to *Ligulalepis* in quotation marks because the specimen discussed in the text (and coded in the analysis) is a braincase, and the material for which the genus was named comprises only scales. In order to be consistent, we have added quotation marks for '*Ligulalepis*' in the text.

Reviewer #2 (Remarks to the Author):

This is a well researched and well written manuscript. Whilst it could be considered just

another description and analysis of a partial fossil, the results from South China in recent decades have proved to be the most significant vertebrate fossil discoveries for the early part of our evolutionary history for perhaps the last century. For some reason that place was an 'evolutionary cradle' for all gnathostomes (jawed vertebrates), with numerous taxa either older than, or more primitive than (or both) comparable fossils from anywhere else in the world.

The new taxon *Ptyctolepis* is yet another example, and like any new fossil, its significance depends on a competent analysis of its morphology in the context of current competing hypotheses. This has been very adequately done in this paper in my opinion, with clear description of new features and character combinations, followed by a full consideration of the implications for interpretation of morphology, and phylogenetic placement, for other key taxa from China, many of which have been described in *Nature*.

Thus there is no problem to accept the major claim of the paper, that it 'illuminates patterns of character evolution in early bony fishes'. It will certainly be of interest to, and influence thinking within, the field of early vertebrate studies.

This paper is also likely to have wider influence, because it concerns the character combinations that define not only various extinct groups, but the two most diverse groups of living vertebrates, the Actinopterygii, dominating the modern aquatic environment, and the ancestral lobe-fins [sarcopterygians], and all their tetrapod descendants in the modern terrestrial environment.

Generally this MS is very well written and clearly expressed. The figures are excellent and necessary, and the MS cannot be shortened without loss of important information. Nevertheless, I suggest the authors consider various points where I have small queries, or can suggest clarification of the text. These are marked on the attached pdf, and listed below. So I would propose perhaps that these minor revisions be done, subject to editorial advice. However none is important enough to detract from the high quality of this submission, which in my assessment would be acceptable for publication in its current form.

Detailed queries/comments [see annotated pdf]:

lines 20-22. perhaps one or other 'character-rich' could be replaced by other words [sounds a bit repetitive]. I suggest some alternative words in line 22.

[Response]: We have revised the text changed the words as suggested to. Now the sentence is: '**Character-rich endocrania are** incompletely preserved for early bony fishes, limiting a detailed understanding of **complex internal morphology and** evolutionary changes in the cranium'. See lines 20-22.

line 23. Other groups [e.g. psarolepids] have been regarded as stem sarcopterygians, even if analysis of this new taxon suggests otherwise, so better not to claim it is 'the first sarcopterygian' in the Abstract.

[Response]: We have revised 'the first stem sarcopterygian' to 'a new sarcopterygian'. See line 22.

lines 61-62. Would be helpful to give more details of size estimates – just a few words in parentheses would do – e.g. other sarcopts about ?? cm total length compared with Ptyctolepis [total length estimated at about ?? cm].

[Response]: It is very difficult to estimate the total length of the specimen from the otoccipital shield, especially given the unusual proportions. As such, we have added the size of the specimen to the text as followed: ‘...measuring with 4.4 cm in width, 1.8 cm in length and 2.7 cm in height, ...’ See lines 60.

lines 103-105. I get the impression from the figures that the holotype specimen may be very slightly distorted (i.e. not completely symmetrical, ignoring missing bits). If so this could slightly affect measurements and proportions - if sedimentary strata have been folded, this can stretch or compress fossils in different axes. Distortion would be evident in all the other fossils from the same locality, but I am not familiar with these. If this is relevant, perhaps a brief comment could be added under ‘Locality and Horizon’.

[Response]: The fish specimens from this horizon are three dimensionally preserved, with very little to no distortion. The specimen described here is very slightly distorted, but most likely due to taphonomic processes rather than folding of the strata. Under ‘Locality and Horizon’, we added “Specimens from this horizon are three dimensionally preserved with little distortion.” See lines 92-93.

lines 109-110. There are always openings in hard tissue between ornamental ridges or tubercles, so not sure if the CT data show evidence that these are ‘pores’ comparable to a pore-canal system.

[Response]: In SI Figure 4c, we show that the pores that open on the surface are connected to pore cavities and continuous with neighboring pores ventrally. As such, they fulfil the definition of a pore-canal system.

lines 159-161. Perhaps two sentences here

[Response]: We have rephrased the sentences as suggested. ‘The cerebellum is well developed, although it appears to lack cerebellar auricles. Swellings situated posterior to the widest part of the hindbrain are associated with the sinus superior rather than the cerebellum, and may have partially accommodated the endolymphatic ducts within the cranial cavity (Fig. 2a).’ See lines 161-164.

lines 197-201. The expression seems a bit cumbersome here. The phrase ‘although ... in some recent studies’ could be a separate sentence at the end, and the term ‘verbal argumentation’ would be better replaced with a few words perhaps expanding what the arguments actually are.

[Response]: We have moved part of this to a separate sentence as suggested: “This arrangement has been suggested through verbal argumentation, but not formal analysis, in some recent studies^{10,18,19}.” See lines 201-202.

line 231. Any idea what may have caused that bias? i.e. preservation bias, collecting bias?

[Response]: We have edited this sentence to ‘as preservation appears biased...’. See line 233.

lines 271-272. I think this is better expressed as an assumption based on outgroup comparisons.

[Response]: We have rephrased the sentence as suggested. “Comparison with outgroups such as actinopterygians (e.g. *Mimipiscis*) and chondrichthyans (e.g. *Cobelodus*) suggests that this is the primitive condition.” See lines 272-273.

lines 312-314. I think more clearly expressed if the phrase ‘contrary to ... analyses’ is moved to the end; and authors to check where the three cited references are best placed.

[Response]: We rephrased the sentence as suggested. ‘..., meaning that dermal bone histology of psarolepids is not compelling evidence of sarcopterygian affinity. These...of discussion that these taxa might be stem osteichthyans rather than early sarcopterygians^{10,18,19}, contrary to the apparent consensus arising from formal analyses.’ See lines 308-310.

Other

Minor typos/corrections/suggested improvements are also marked in lines 66, 132, 148, 211, 212, 242, 254, 324, 328, 331, 339, 341, 342, 348.

[Response]: We modified all of them as suggested. See below.

line 66 – 66 sarcopterygian.^ST
^

[Response]: Modified. See line 65.

line 132 – 132 Posteriorly, the transverse otic process (cf. *Chetrolepis*⁴²).

[Response]: Modified. See line 132.

line 148 – 148 trunk musculature, but is not divided into distinct regions (cf. *Youngolepis*).

[Response]: Modified. See line 150.

line 211 – 211 areas of the tree are:
our

[Response]: Modified. See line 212.

line 212 – 212 ‘acanthodians’ have collapsed into a series of polytomies. The ~~large~~^{high} number

[Response]: Modified. See line 214.

line 242 – ²⁴² with a macromeric dermal skeleton lies close

[Response]: Modified. See line 242.

line 254 – ²⁵⁴ ~~such as~~ *Psycotolepis* (Fig. 3b). The fontanelle

[Response]: Modified. ‘... based on our new evidence from...’. See line 254.

line 324 – ³²⁴ optimized multiple in

[Response]: Modified. See line 323.

line 328 – ³²⁸ absent only in the anatomically peculiar *Dialipina*. In addition to these

[Response]: Modified. See line 329.

line 331 – ³³¹ suspected. Rather than ~~representing~~ a synapomorph

[Response]: Modified. See line 331.

line 339 – ³³⁹ between this solution and one that places these taxa in a more conventional position on

[Response]: Modified. See line 340.

line 341 – ³⁴¹ discoveries of early osteichthyan material from the Silurian of China and additi

[Response]: Modified. See line 342.

line 342 – ³⁴² study of existing fossils will help to clarify t

[Response]: Modified. See line 342.

line 348 – ³⁴⁸ present ambiguities, we regard the placement of these taxa as uncertain but

[Response]: Modified. See line 348.

Reviewer #3 (Remarks to the Author):

The authors describe a partial braincase of a new Early Devonian osteichthyan. Although partial, the material is well preserved and allows recovery of characters related to the oto-occipital skeleton as well as an endocast of the hind brain and ears. The authors specifically identify several features that have bearing on the phylogenetic placement of this new taxon and implications for basal osteichthyan evolution more generally. One key interpretation is that the new taxon is a stem sarcopterygian based

on three characters shared with crown sarcopterygians: 1) spiracular groove absent from lateral commissure; 2) double hyoid arch articulation on braincase; and 3) parachordals mediolaterally constricted relative to otic capsules. The most important conclusions of the manuscript relate to the new tree presented in Figure 3, which differs from previous interpretations in formally placing forms such as †Guiyu and †Psarolepis as stem osteichthyans, †Meemania as a stem actinopterygian, and the new taxon as a stem sarcopterygian. If this interpretation holds up, it will substantially contribute to our interpretation of basal osteichthyan evolution.

My comments on the manuscript are minor and editorial, but I would like to suggest two style points that would improve general readability of this (and other) papers in paleontology published in Nature Communications. First, I am a strong proponent of the dagger (†) symbol because it instantly clarifies to readers that the authors are referring to an extinct taxon, about which we will always know less than an extant taxon. It also helps clarify which named higher groups are extant, and this is particularly important in any tree that combines extinct and extant taxa as in Figure 3.

[Response]: While we agree with the reviewer that a dagger symbol can be a useful tool, unfortunately this is not the house style of Nature journals and we are unable to include it.

Second, in figure callouts that refer to specific anatomical structures, it is a great help to readers if the abbreviation to the feature can be included as part of the callout. In reviewing this manuscript, I found several instances where an anatomical feature mentioned in a callout was difficult to find on the figure itself, and note some instances of this in my detailed comments below. Now I realize that my quibbles about these two style points may trace to specific style preferences of Nature Communications; but if so, then I would like to see a change in the journal's style standards to accommodate these preferences at an author's discretion.

[Response]: As with the previous comment, this is not the house style of Nature journals, and we are unable to deviate from editorial standards.

Line 20 – “this dominant vertebrate group” antecedent to “this” could be unclear; also, what exactly do you mean by “dominant?” Presumably speciose – but this is so self evident that I am not sure this phrase helps your story.

[Response]: We have rephrased this to ‘in this major jawed vertebrate group’. See lines 19-20.

Line 20 and Line 21 – Use of the phrase “character-rich endocrania” twice in two lines reads like a hard sell. Once would be enough.

[Response]: We have revised this sentence in response to this and Reviewer 2's comments: Modified. Now the sentence is: ‘**Character-rich endocrania are** incompletely preserved for early bony fishes, limiting a detailed understanding of **complex internal morphology and** evolutionary changes in the cranium’. See lines 20-22.

Line 46 – Perhaps replace “primitive” with “plesiomorphic”

[Response]: Modified. See line 45.

Line 59 – Replace “This taxon is only represented” with “This taxon is represented”

[Response]: Modified. See line 58.

Line 66 – Replace “sarcopterygian” with “sarcopterygians”

[Response]: Modified. See line 65.

Line 67 – Replace “while the braincase shows an absence of vestibular fontanelles and an unprecedented condition of the hyomandibular facet” with “while the braincase lacks vestibular fontanelles and shows an unprecedented condition of the hyomandibular facet.”

[Response]: Modified. See line 67-68.

Line 84 – Replace “a complete posterior cranial portion” with “a complete posterior cranial portion of the skull”

[Response]: Modified. See line 84.

Lines 102-103 – Replace “The skull roof is represented only by the posterior cranial portion” with “The skull roof is represented by the posterior cranial portion of the skull”

[Response]: Modified. See line 103.

Line 120 – Its the callout to Figure 2 correct here? Does not seem relevant to the rest of the paragraph. Perhaps this should be Figure 1?

[Response]: Corrected. See line 120.

Lines 139-140 – This is an example where an indication to a specific label in the figure would help: “The otoccipital fissure, through which the vagus nerve exits, is well developed (Fig. 1e).” Where exactly is this in the figure?

[Response]: We have added a label for the exit of the vague nerve to figure 1. We are unable to refer to specific abbreviations in the text, but abbreviations will be displayed underneath the figure in accordance with house style.

Lines 145-146 – Again, I could not easily see that “articular area for the first suprapharyngobranchial is marked by a stout post-otic process at the level of the foramen magnum (Fig. 1c–e).” A callout to a specific abbreviation would have helped.

[Response]: As noted above, we are unable to include specific anatomical abbreviations in the text, and abbreviations will be provided immediately underneath the figure. We have rephrased the text slightly as follows: ‘The articular area for the first suprapharyngobranchial is **borne on** a stout post-otic process at the level of the foramen magnum’. See line 148.

Line 172 – Is the callout to Figure 1 correct? Perhaps Figure 2 shows this better?

[Response]: Corrected. See line 174.

Line 224 – Authors state that “intact braincases for members of the group are rarely completely preserved.” The sentence seems to imply that the new specimen has an intact braincase, but this specimen is not an intact braincase either, just the oto-occipital region.

[Response]: As we clarify in the following sentences that the otoccipital is particularly poorly represented, we retain our original phrasing here. But for clarity, we added a few words in the abstract ‘...a completely ossified otoccipital **division of the braincase**’. See line 24.